# Understanding the Loss Surface of Single-Layered Neural Networks for Binary Classification

**Shiyu Liang, Ruoyu Sun, R. Srikant**
University of Illinois at Urbana-Champaign
{sliang26, ruoyus, rsrikant}@illinois.edu

**Yixuan Li**
Cornell University
yl2363@cornell.edu

## Abstract

It is widely conjectured that the reason that training algorithms for neural networks are successful because all local minima lead to similar performance; for example, see (LeCun et al., 2015; Choromanska et al., 2015; Dauphin et al., 2014). Performance is typically measured in terms of two metrics: training performance and generalization performance. Here we focus on the training performance of single-layered neural networks for binary classification, and provide conditions under which the training error is zero at all local minima of a smooth hinge loss function. Our conditions are roughly in the following form: the neurons have to be strictly convex and the surrogate loss function should be a smooth version of hinge loss. We also provide counterexamples to show that when the loss function is replaced with quadratic loss or logistic loss, the result may not hold.

## 1 Introduction

Local search algorithms like stochastic gradient descent (Bottou, 2010) or variants have gained huge success in training deep neural networks (see, (Krizhevsky et al., 2012; Goodfellow et al., 2013; Wan et al., 2013), for example). Despite the spurious saddle points and local minima on the loss surface (Dauphin et al., 2014), it has been widely conjectured that all local minima of the empirical loss lead to similar training performance (LeCun et al., 2015; Choromanska et al., 2015).

In the setting of regression problems, theoretical justifications (Andoni et al., 2014; Sedghi & Anandkumar, 2014; Janzamin et al., 2015; Haeffele & Vidal, 2015; Gautier et al., 2016; Brutzkus & Globerson, 2017; Soltanolkotabi, 2017; Soudry & Hoffer, 2017; Goel & Klivans, 2017; Du et al., 2017; Zhong et al., 2017; Li & Yuan, 2017; Baldi & Hornik, 1989; Kawaguchi, 2016; Freeman & Bruna, 2016; Hardt & Ma, 2017; Yun et al., 2017; Nguyen & Hein, 2017a;b) has been established to support the conjecture that all local minima lead to similar training performance. Although the loss surfaces in regression tasks have been well studied, the theoretical understanding of loss surfaces in classification tasks is still limited. Nguyen & Hein (2017b), Boob & Lan (2017) and Soltanolkotabi et al. (2017) treat the classification problem as the regression problem by using quadratic loss, and show that (almost) all local minima are global minima. However, the global minimum of the quadratic loss does not necessarily have zero misclassification error even in the simplest cases (e.g., every global minimum of quadratic loss can have non-zero misclassification error even when the dataset is linearly separable and the network is a linear network). This issue was mentioned in (Nguyen & Hein, 2017a) and a different loss function was used, but their result only studied the linearly separable case and a subset of the critical points.

In view of the prior work, the context and contributions of our paper are as follows:

- Prior work on quadratic and related loss functions suggest that one can achieve zero misclassification error at all local minima by overparameterizing the neural network. The reason for over-parameterization is that the quadratic loss function tries to match the output of the neural network to the label of each training sample.

- On the other hand, hinge loss-type functions only try to match the sign of the outputs with the labels. So it may be possible to achieve zero misclassification error without over-

parametrization. We provide conditions under which the misclassification error of single layered neural networks is zero at all local minima for hinge-loss functions.

- Our conditions are roughly in the following form: the neurons have to be strictly convex and the surrogate loss function is a smooth version of the hinge loss function.

- We provide counterexamples to show that when the loss function is replaced with quadratic loss or logistic loss, the result may not hold.

- We establish our results under the assumption that either the dataset is linearly separable or the positively and negatively labeled samples are located on different subspaces. Whether this assumption is necessary is an open problem.

The outline of this paper is as follows. In Section 2, we present the necessary definitions. In Section 3, we present the main results and discuss the impact of loss functions on the main results. Conclusions are presented in Section 4. A longer version of this paper will appear on arXiv.

## 2 PRELIMINARIES

**Network models.** Given an input $x$ of dimension $d$, we consider a single layered network with a single output for binary classification, i.e., $f(x; \boldsymbol{\theta}) = a_0 + \boldsymbol{a}^\top \sigma(\boldsymbol{W}^\top x)$, where scalar $a_0$ denotes the bias, vector $\boldsymbol{a} \in \mathbb{R}^M$ denotes the weight vector, $\boldsymbol{W} \in \mathbb{R}^{d \times M}$ denotes the weight matrix, integer $M$ denotes the number of neurons and vector $\boldsymbol{\theta}$ denotes the parameterization of the network. We make the following assumption on the neural activation function.

**Assumption 1** *Assume that neurons $\sigma$ in the network $f$ are real analytic and satisfy $\sigma''(z) > 0$ for all $z \in \mathbb{R}$.*

Here, we list a few neurons which can be used in the network: softplus neuron, i.e., $\sigma(z) = \log_2(1 + e^z)$, quadratic neuron, i.e, $\sigma(z) = z^2$, etc.

**Data distribution.** In this paper, we consider binary classification tasks where each sample $(\boldsymbol{X}, Y) \in \mathbb{R}^d \times \{-1, 1\}$ is drawn from an underlying data distribution $\mathbb{P}_{\boldsymbol{X} \times Y}$ defined on $\mathbb{R}^d \times \{-1, 1\}$. The sample $(\boldsymbol{X}, Y)$ is considered positive if $Y = 1$, and negative otherwise. Let $\mathcal{E} = \{\boldsymbol{e}_1, ..., \boldsymbol{e}_d\}$ denote a set of orthonormal basis on the space $\mathbb{R}^d$. Let $\mathcal{U}_+$ and $\mathcal{U}_-$ denote two subsets of $\mathcal{E}$ such that all positive and negative samples are located on the linear span of the set $\mathcal{U}_+$ and $\mathcal{U}_-$, respectively, i.e., $\mathbb{P}_{\boldsymbol{X}|Y}(\boldsymbol{X} \in \text{Span}(\mathcal{U}_+)|Y = 1) = 1$ and $\mathbb{P}_{\boldsymbol{X}|Y}(\boldsymbol{X} \in \text{Span}(\mathcal{U}_-)|Y = -1) = 1$. Let $r$ denote the size of the set $\mathcal{U}_+ \cup \mathcal{U}_-$, $r_+$ denote the size of the set $\mathcal{U}_+$ and $r_-$ denote the size of the set $\mathcal{U}_-$, respectively.

**Assumption 2** *Assume that for random vectors $\boldsymbol{X}_1, ..., \boldsymbol{X}_{r_+}$ independently drawn from the distribution $\mathbb{P}_{\boldsymbol{X}|Y=1}$ and $\boldsymbol{Z}_1, ..., \boldsymbol{Z}_{r_-}$ independently drawn from the distribution $\mathbb{P}_{\boldsymbol{X}|Y=-1}$, matrices $(\boldsymbol{X}_1, ..., \boldsymbol{X}_{r_+}) \in \mathbb{R}^{r_+ \times d}$ and $(\boldsymbol{Z}_1, ..., \boldsymbol{Z}_{r_-}) \in \mathbb{R}^{r_- \times d}$ are full rank matrices with probability one.*

Assumption 2 states that support of the conditional distribution $\mathbb{P}_{\boldsymbol{X}|Y=1}$ is sufficiently rich so that $r_+$ samples drawn from it will be linearly independent. In other words, by stating this assumption, we are avoiding trivial cases where all the positively labeled points are located in a very small subset of the linear span of $\mathcal{U}_+$. Similarly for the negatively labeled samples.

**Assumption 3** *Assume $|\mathcal{U}_+ \cup \mathcal{U}_-| > \max\{|\mathcal{U}_+|, |\mathcal{U}_-|\}$, i.e., $r > \max\{r_+, r_-\}$.*

Assumption 3 assumes that the positive and negative samples are not located on the same linear subspace. Previous works (Belhumeur et al., 1997; Chennubhotla & Jepson, 2001; Cootes et al., 2001; Belhumeur et al., 1997) have observed that some classes of natural images (e.g., images of faces, handwritten digits, etc) can be reconstructed from lower-dimensional representations. For example, using dimensionality reduction methods such as PCA, one can approximately reconstruct the original image from only a small number of principal components (Belhumeur et al., 1997; Chennubhotla & Jepson, 2001). Here, Assumption 3 states that both the positively and negatively labeled samples have lower-dimensional representations, and they do not exist in the same lower-dimensional subspace.

**Loss and error.** Let $\mathcal{D} = \{(x_i, y_i)\}_{i=1}^n$ denote a dataset with $n$ samples, each independently drawn from the distribution $\mathbb{P}_{\boldsymbol{X} \times Y}$. In this paper, we consider the following loss function.

**Assumption 4** *Assume that the loss function is $\ell_p(z) = [\max\{z + 1, 0\}]^{p+1}$, where $p \in \mathbb{N}$.*

Given a neural network $f(x; \boldsymbol{\theta})$ parameterized by $\boldsymbol{\theta}$, in binary classification tasks, we define the **empirical loss** $\hat{L}_n(\boldsymbol{\theta})$ as the average loss of the network $f$ on a sample in the dataset $\mathcal{D}$, i.e., $\hat{L}_n(\boldsymbol{\theta}; p) = \frac{1}{n} \sum_{i=1}^n \ell_p(-y_i f(x_i; \boldsymbol{\theta}))$. We define the **training error** (also called the **misclassification error**) $\hat{R}_n(\boldsymbol{\theta})$ as the misclassification rate of the neural network $f(x; \boldsymbol{\theta})$ on the dataset $\mathcal{D}$, i.e., $\hat{R}_n(\boldsymbol{\theta}) = \frac{1}{n} \sum_{i=1}^n \mathbb{I}\{y_i \neq \operatorname{sgn}(f(x_i; \boldsymbol{\theta}))\}$, where $\mathbb{I}\{\cdot\}$ is the indicator function and sgn is the sign function, i.e., $\operatorname{sgn}(z) = 1$ if $z \geq 0$ and $\operatorname{sgn}(z) = -1$, otherwise.

## 3 MAIN RESULTS AND DISCUSSIONS

In this section, we present the following theorem to show that when assumptions 1-4 are satisfied, every local minimum of the empirical loss function has zero training error if the number of neurons in the network $f_S$ are chosen appropriately.

**Theorem 1** *Suppose that assumptions 1-4 are satisfied. Assume that samples in the dataset $\mathcal{D} = \{(x_i, y_i)\}_{i=1}^n, n \geq 1$ are independently drawn from the distribution $\mathbb{P}_{\boldsymbol{X} \times Y}$. Assume that the number of neurons $M$ in the network $f$ satisfies $M \geq 2 \max\{\frac{n}{\Delta r}, r_+, r_-\}$, where $\Delta r = r - \max\{r_+, r_-\}$. If $\boldsymbol{\theta}^*$ is a local minimum of the loss function $\hat{L}_n(\boldsymbol{\theta}; p)$ and $p \geq 6$, then $\hat{R}_n(\boldsymbol{\theta}^*) = 0$ holds with probability one.*

**Remark:** The positiveness of $\Delta r$ is guaranteed by Assumption 3. In the worst case (e.g., $\Delta r = 1$ and $\Delta r = 2$), the number of neurons needs to be at least greater than the number of samples, i.e., $M \geq n$. However, when the two orthonormal basis sets $\mathcal{U}_+$ and $\mathcal{U}_-$ differ significantly (i.e., $\Delta r \gg 1$), the number of neurons required by Theorem 1 can be significantly smaller than the number of samples (i.e., $n \gg 2n/\Delta r$). In fact, we can show that, when the neuron has quadratic activation function $\sigma(z) = z^2$, the assumption $M \geq 2n/\Delta r$ can be further relaxed such that the number of neurons is independent of the number of samples.

**Quadratic loss.** The quadratic loss $\ell(z) = (1 - z)^2$ has been well-studied in prior works. It has been shown that when the loss function is quadratic, under certain assumptions, all local minima of the empirical loss are global minima. However, the global minimum of the quadratic loss does not necessarily have zero misclassification error, even in the realizable case (i.e., the case where there exists a set of parameters such that the network achieves zero misclassification error on the dataset or the data distribution). To illustrate this, we provide an example in Appendix A and show that, when the loss function is replaced with quadratic loss, even if the other conditions in Theorem 1 are satisfied, every global minimum of the empirical loss has a training error larger than $1/8$ with a positive probability. In other words, our main results do hold for the quadratic loss.

**Logistic loss.** The logistic loss $\ell(z) = \log_2(1 + e^z)$ is different from the loss functions conditioned in Assumption 4, since the logistic loss does not have a global minimum on $\mathbb{R}$. Here, for the logistic loss function, we show that even if the remaining assumptions in Theorem 1 hold, every critical point is a saddle point. In other words, Theorem 1 does not hold for logistic loss.

**Proposition 1** *Assume that the loss function is the logistic loss, i.e., $\ell(z) = \log_2(1 + e^z)$. Assume that assumptions 2-1 are satisfied. Assume that samples in the dataset $\mathcal{D} = \{(x_i, y_i)\}_{i=1}^n, n \geq 1$ are independently drawn from the distribution $\mathbb{P}_{\boldsymbol{X} \times Y}$. Assume that the number of neurons $M$ in the network $f$ satisfies $M \geq 2 \max\{\frac{n}{\Delta r}, r_+, r_-\}$, where $\Delta r = r - \max\{r_+, r_-\}$. If $\boldsymbol{\theta}^*$ denotes a critical point of the empirical loss $\hat{L}_n(\boldsymbol{\theta})$, then $\boldsymbol{\theta}^*$ is a saddle point. In particular, there are no local minima.*

In addition, we note here that when the neurons are replaced with rectified linear units (ReLUs), leaky rectified linear units (Leaky-ReLU) and sigmoid neurons, Theorem 1 does not hold.

## 4 CONCLUSIONS

In this paper, we studied the surface of a smooth version of the hinge loss function in binary classification problems. We provided conditions under which the neural network has zero misclassification error at all local minima and also provide counterexamples to show that when the loss function is replaced with quadratic loss or logistic loss, the result may not hold. Further work involves exploiting our results to design efficient training algorithms classification tasks using neural networks.

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

## A EXAMPLE

**Example 1** *Let the distribution $\mathbb{P}_{\boldsymbol{X} \times Y}$ defined on $\mathbb{R}^2 \times \{-1, 1\}$ satisfy that $\mathbb{P}(Y = 1) = \mathbb{P}(Y = -1) = 0.5$, $\mathbb{P}(X = (\alpha, 0)|Y = 1) = \mathbb{P}(X = (1, 0)|Y = 1) = 0.5$ and $\mathbb{P}(X = (0, \alpha)|Y = -1) = \mathbb{P}(X = (0, 1)|Y = -1) = 0.5$. Assume that samples in the dataset $\mathcal{D} = \{(x_i, y_i)\}_{i=1}^{4n}$ are independently drawn from the distribution $\mathbb{P}_{\boldsymbol{X} \times Y}$. Assume that the single layered network $f$ has $M \geq 2$ neurons and all neurons in the network $f$ are quadratic neurons, i.e., $\sigma(z) = z^2$. Then there exists an $\alpha \in [0, 1]$ such that every global minimum of the empirical loss function $\hat{L}_{4n}(\boldsymbol{\theta}) = \frac{1}{4n} \sum_{i=1}^{4n} (1 - y_i f(x_i; \boldsymbol{\theta}))^2$ has a training error greater than $1/8$ with probability at least $\Omega(1/n^3)$.*

**Remark:** This is a counterexample for Theorem 1. It is easy to check that the distribution satisfies assumption 2 and 3, where $r = 2 > \max\{1, 1\} = \max\{r_+, r_-\}$.

