# OpenReview forum: "Understanding the Loss Surface of Single-Layered Neural Networks for Binary Classification"
_ICLR.cc/2018/Workshop — Accept_

### Official Review · AnonReviewer1 · 2018-02-28
**Strong and Solid Theory Paper**

**Rating:** 9
**Confidence:** 4

**Review:**

This is a very strong and solid theory paper on neural networks. Analyzing classification is often a more difficult problem than regression. This paper considers many aspects that affect the classification error, including: loss function, activation function and data distribution. The data distribution assumption is very interesting and may inspire further research. Further, this paper also provides counter examples which are useful for theoretical researchers.
Overall I recommend to accept strongly.

(Since proofs are not included, I cannot check the correctness.)

---

### Official Review · AnonReviewer3 · 2018-03-04
**Interesting progress in the classification setting**

**Rating:** 7
**Confidence:** 4

**Review:**

The paper has some nice results on the loss landscape of a single layer neural network when the activation function is strongly convex and when the loss function is a smooth version of the hinge loss then every local minimal is the same as the global minimum.

The paper is clearly written in that it explicitly lists out the four assumptions under which the conditions hold. The result is also significant in that many recent works focus on the regression setting.

It seems the work replies heavily on the smoothness and large curvature of the functions, for example, the activation functions need to be strictly convex and the smoothed hinge loss must satisfy p >= 6. It will be interesting to talk about briefly why such smoothness is needed. And what is the difficulty with proving ReLU and leaky ReLU?

---

### Official Review · AnonReviewer2 · 2018-03-10
**Review for "Understanding the Loss Surface of Single-Layered Neural Networks for Binary Classification"**

**Rating:** 4
**Confidence:** 4

**Review:**

This paper shows that under certain conditions, all local minima of a single-layer neural network has zero training error for binary classification. The key assumption is on the distribution of input data. It assumes that the positive examples and the negative examples belong to two different low-dimensional linear subspaces, and these two subspaces don't overlap.

Comparing to existing work, the result of this paper is interesting in that it doesn't require the number of hidden nodes to be more than the number of training examples. In particular, when the subspaces of positive and negative examples are highly separable, the number of hidden nodes can be much smaller than the number of samples. Nevertheless, the conclusion only holds for certain smooth activation functions and loss.

The main limitation is the strong assumption on the data distribution, which never holds in practice. Indeed, under this paper's assumption, we can fit two PCA models, on the positive examples and the negative examples respectively, then for any data point, we can use the fitting error on these two models to predict its binary class (the correct class has zero fitting error, while the incorrect class has strictly positive fitting error). This simple approach can achieve zero training and testing error with much lower sample complexity, by training much less parameters. It means that the problem itself is so easy that neural network is a complete overkill. The theoretical result for such a problem is thus not interesting, especially when it only holds under restrictive assumptions.

---

### Decision · Program_Chairs · 2018-03-20
**ICLR 2018 Workshop Acceptance Decision**

**Decision:**

Accept

**Comment:**

Congratulations, your paper was accepted to the ICLR workshop.